# Using a Novel Floating Grinding Process to Improve the Surface Roughness Parameter of a Magnetic Head

**DOI:** 10.3390/nano12162763

**Published:** 2022-08-12

**Authors:** Xionghua Jiang

**Affiliations:** Department of Metal Materials, College of Materials Science and Engineering, Dongguan University of Technology, Dongguan 523000, China; 2016055@dgut.edu.cn

**Keywords:** magnetic head, fine grinding, black spot

## Abstract

This work concentrated on the improvement of the surface roughness of a magnetic head, through the use of an ultrafine nanodiamond slurry, and a novel floating grinding process, which optimize different experimental factors required for the fine grinding of a magnetic head. The preparation of the grinding plate was confirmed by the observation of the surface change, depth detection, and flatness after ultrafine nanodiamonds were embedded into it by a Keyence high-power microscope at a 20 K magnification. The flatness was measured by a TOTO instrument. The optimum conditions were found to be a pit ratio reach of 30:70 and a plate flatness (average) of 1.8 μm. The rotation speed and vibration frequency were 0.3 and 10 rpm, respectively, for the grinding process. The morphology, size, and elemental composition of blackspots were investigated by SEM, AES, AFM, and transmission electron microscopy (TEM) analysis, which showed that the diameter of the diamonds in the slurry was important for grinding surface improvement. A novel method was proposed in this study to fine grind a magnetic head using a small-sized diamond slurry (100 nm) in conjunction with a novel float lapping method. Comparison experiments were performed under both normal conditions and improved conditions. The results show that by using the novel float lapping method with a small-sized diamond slurry, the minimum roughness was obtained. The finest roughness obtained for the slider surface reached 0.165 nm without blackspots or scratches.

## 1. Introduction

The magnetic head is a key component of a hard disk: the read/write capability and precision of the magnetic head directly affect the storage density and service life of the hard disk for big data storage. With the increasing storage density of perpendicular magnetic recording and the gradual maturity of thermally assisted magnetic storage technology with increased storage capacity, the magnetic storage unit has sharply shrunk in size. The surface accuracy requirements of signal read-write heads have become increasingly stringent, even reaching the limit of existing grinding machines. Therefore, a novel ultra-precision grinding mechanism and polishing technology must be urgently developed.

Figure 1 shows the precision grinding surface of a magnetic head, which consists of the simultaneous grinding of the matrix of the magnetic head, the metallic magnetic pole tip (located on the floating slider surface) and an Al_2_O_3_ protective layer; a prescribed quantity of material must be removed. In addition, surface defects in the magnetic head (including roughness, black spots, scratches, pole tip recession, etc.) must be controlled [1].

The quantity of metal removed at the pole tip must be controlled at the nanometer level: compared with other materials, the metallic material of the magnetic head has a considerably more complex structure and more easily oxidizes in air, which makes it significantly more difficult to grind the magnetic head. Few studies have been performed to date on controlling the surface topography of the pole tip of the magnetic head, and there are even fewer reports on the grinding mechanism of the composite material of the magnetic head. Several scientists [2,3,4,5] have used traditional grinding technology to study the physical and mechanical aspects of the polishing method of the magnetic head, grinding disc, grinding ring and polishing pad and have proposed a kinematic equation for slider grinding and machine polishing. Studies have been performed using lubricating oil (grinding oil) to investigate the thermal control of the flying height of a magnetic head [6,7,8]. Other studies have shown that lubricating oil molecules migrate from the magnetic disk to the magnetic head when these components are not in contact. Smooth regions in the magnetic head frequently occur. Fe_2_O_3_ and SiO_2_ nonporous composite abrasives have been developed and used to grind the head arm of a hard disk, which results in good chemical and mechanical properties for the hard disk [9,10].

Domestic and foreign researchers have performed extensive research on different materials for use in ultra-precision grinding [11,12,13,14,15]. In 2020, Deng Hui’s team from SUS Tech [16] proposed a polishing technology based on the isotropic etching of profile envelopes. By optimizing parameters such as the type of electrolyte and breakdown voltage, isotropic etching was achieved on a titanium surface. Etching and ultra-precision polishing resulted in a polishing efficiency of up to 15.1 m/min, where 3 min of polishing caused the surface roughness to rapidly decrease from 64.1 to 1.23 nm.

Prominent differences in the physical and chemical properties of the constituent materials in a magnetic head make it difficult to apply ultra-precision grinding to a multimaterial composite device, and it is difficult for device specifications to be satisfied using conventional mechanical and chemical grinding methods. Therefore, a novel float grinding process was developed in this study, where an ultrafine nanodiamond slurry (100 nm) is used for precision grinding. A Keyence high-resolution microscope was used to observe the surface topography of a tin plate embedded with nanodiamonds and determine the embedded depth, and a dynamic model of the shaving process of the grinding ring was built to analyze the grinding process. The surface morphology of the grinding plate was characterized by a Keyence high-power microscope at a 20 K magnification, and the flatness of the plate was measured by a TOTO instrument. Black spots on the magnetic head were observed using SEM, XRD, AES, and TEM. Finally, AFM and SEM were performed to compare the surface characteristics of the plate and pole tip under normal and improved conditions (i.e., using fine grinding).

## 2. Materials and Methods

### 2.1. Preparation and Microstructure Characterization of the Grinding Plate

The fabrication of a grinding disc consists of turning, iron-ring shaving, and porcelain-ring shaving processes. To ensure that there is a sufficient quantity of nanodiamond abrasive for the cutting process, nanodiamond particles are continuously brought into the cutting area during the grinding process, and the chips generated during the grinding process are removed in a timely manner. The surface of the grinding plate must be turned into a serrated shape to facilitate cooling. A EJW-460SFN-AR lathe (Engis Company, Tokyo, Japan) was used in the experiments in this study. A rough plate was selected for the grinding plate with an outer diameter of 400 mm, and the plate flatness was controlled to within 2 μm. The best condition was found to be 1.8 μm.

The turning process consists of (1) checking and cleaning the surface of the plate, (2) setting the parameters of the plate and (3) selecting the specification of the turning tool. The turning tool is installed, and the plate is turned. The flatness obtained by turning can be controlled up to 0.5 μm/400 mm, which requires a pit width of 100 μm. Cutters with different angles, 30-degree single-crystal cutters and 45-degree polycrystalline cutters, respectively, are used to modify the pit ratio. The parameters were set: depth 5 μm, feed rate 20 mm/min, plate speed 200 rpm. During the shaving process, an iron ring (without a load) is employed for the first 3–5 min. The plate surface is checked for abnormalities, and the abrasive distribution status of the nanodiamonds embedded in the tin plate is shown in Figure 2a. An 8 kg load is added and shaving is continued for 10 more minutes. As shown in Figure 2b, a 100× microscope was used to check whether the pit pattern meets the requirements. The optimized pit ratio was found to reach 30:70, and the control range of the plate was 30 ± 10 μm. The pit ratio can also be explained by the land groove ratio. The pit (or the groove) is similar to a backflow groove, which takes the cutting fragments away through the grinding oil. It can also store grinding oil to ensure lubrication. The land is in contact with the magnetic stripe (row bar), imbedded with abrasive particles (nanodiamonds), and plays a grinding role in the lapping process.

The shaving process of the iron ring involves processing the serrated peaks on the plate surface into a platform, where the platform roughness is the main factor that affects the quality of the grinding surface. During the grinding process, the grooves are filled with grinding slurry to provide abrasive diamond grains for grinding, and the grinding chips are discharged through the grooves to reduce the heat generated during processing and maintain a stable grinding temperature.

After checking that the pit pattern ratio is normal, the trimming process of the porcelain ring is performed. A layer of nanodiamond abrasive is evenly sprayed on the plate surface, a clean ceramic ring is placed at the center of the plate surface, and a 16 kg load is added to press the nanodiamond particles into the plate. White oil is used to clean the center and side edges of the plate during grinding. The scheme of the shaving process of the grinding ring is shown in Figure 3.

Ultrasonication is used to ensure that the nanodiamond particles are pressed uniformly into the plate. First, adjustable piezoelectric ceramics are added to the ceramic ring; then, an alternating voltage of frequency of ~20 kHz is applied to the piezoelectric ceramics, and the voltage frequency is adjusted to the frequency of the ceramic ring. The voltage amplitude is adjusted by adjusting the magnitude of the voltage.

After finishing, the surface morphology of the plate is characterized by a Keyence high-power microscope at a 20 K magnification, and the flatness is measured by a TOTO flatness measurement instrument (TOTO Inc., Kitakyushu, Japan). The results are shown in Figure 4.

### 2.2. Ultra-Precision Grinding Experiments

The ultraprecision grinding process (Figure 5) consists of two stages: in the first stage, the pressure is 30.4 N, the rotation speed of the plate is set to 3 rpm, and the swing frequency of the loading plate is 3 times/min. In the second stage, the same briquette weight is used, and the rotation speed and vibration frequency are 0.3 and 10 rpm, respectively (with no delay time). A prescribed weight is placed on the loading plate, and the surface is sprayed with grinding oil.

The traditional grinding process is shown in Figure 6a. There are no lands or grooves in the plate, and the float grinding process is developed with land and grooves as shown in Figure 6b. The nanodiamond cutting depth is not controllable under either traditional or float grinding processes, which makes it easy for many floating diamonds to produce surface machining defects in the head, such as black spots and scratches. Figure 6c shows the novel machining method developed in this study: nanodiamonds are embedded in the plate, lubricant is only sprayed on the plate during grinding, and the fixed diamond particles form a mosaic on the surface of the plate, act as a grinding tool, and implement cutting and grinding. This novel method ensures that the diamond particles pressed into the plate protrude at the same height. A micro damage model [17,18] is constructed to grind a hard, brittle material based on indentation fracture mechanics, i.e., the effect of dynamic fracture on the toughness, grinding speed, grinding depth, processing load and material mechanics performance of the grinding surface are considered. This prediction model for the surface microscopic damage is used to control the cutting depth. Parameter optimization is performed to maintain the head below the critical cutting depth of the substrate material and eliminate crack extension and brittleness. Ultrasonically enhanced intelligent control technology is applied to evenly press the nanodiamond particles into the plate, which prevents material spalling, pitting and long cracks and forms a supersmooth surface.

## 3. Results and Discussion

The grinding process of a material is influenced by the interaction of the material properties, abrasive geometry, abrasive plunge motion, and mechanical and thermal loads acting on the workpiece and abrasive particles. The hard, brittle AlTiC ceramic matrix and Al_2_O_3_ protective layer are subjected to a process to eliminate brittleness and plasticity, which involves the removal and peeling of material. Soft and tough metallic materials mainly suffer from deformation wear and cutting wear, which correspond to an erosion wear mechanism. Prominent differences in the corrosion potential and corrosion rate of the constituents during chemical etching and polishing prevent the realization of ultra-precision grinding. We further analyze the ultra-precision grinding of a single material to elucidate the grinding mechanism of a multicomponent composite material, determine guidelines that are widely applicable to grinding methods for various types of materials, and develop a suitable method for grinding magnetic heads.

Several analyses were performed to elucidate the control mechanism for the slider surface. Figure 7 shows a secondary electron image collected by a JSM 6301F SEM of the pole area of the magnetic head. Black spots can be observed on the head pole in the SEM image. Area 3, which is marked in Figure 7, was selected for topography analysis by atomic force microscopy (AFM). The AFM results (Figure 8) show that the black spots are protrusions in the pole area. The protrusion is less than 0.1 nm in diameter and approximately 5 nm in height.

To analyze the composition of the black spots, AES was performed on both normal area and black spots (Points 1 and 2), respectively. Ar+ ion sputtering was used to remove hydrocarbons absorbed on the surface before performing the AES analysis. Figure 9 shows the AES results obtained using a 10 kV primary electron beam and a 10 nA current. Only the substrate components, Ni and Fe, were observed on the normal area. However, a large quantity of carbon was detected on the black spots in addition to the substrate materials. The AES fine spectra of different types of carbon are shown in Figure 10. Spectra 1, 2 and 3 were collected for samples of the black spots, diamond-like carbon (DLC) and surface-contaminated hydrocarbons, respectively. Differences in the peak shape and kinetic energy among the three spectra can be observed. These results imply that the C on the black spots is not DLC or a hydrocarbon. The peaks in Spectrum 1 of the black spots are considerably sharper than those of the other two spectra. In particular, the well-defined plasmon-loss feature in the low-energy regime of Spectrum 1 is very different from the features of the other two spectra. It is well known that diamond-like carbon (DLC) is composed of mixed sp3 and sp2 domains of carbon and hydrogen. The composition of the surface-contaminated hydrocarbons is even more complex. Therefore, the black spot is very likely composed of pure carbon with a well-ordered structure. Figure 11 shows a super-SEM image of a black spot taken with a Hitachi-S5200 scanning electron microscope (spatial resolution ~ 0.5 nm). Figure 12 shows that nanoscale diamond crystals with a perfect structure are embedded into the substrate (an Fe and Ni alloy) at the black spots. The crystal size is less than 40 nm. The black spots are clearly caused by the embedment of nanodiamond grains from the slurry into the magnetic head surface during the lapping process.

The furrow effect can explain the mechanics of grinding with nanodiamonds. Diamond is generally irregular in shape and can have sharp corners. According to Hertz contact theory, the stress at a sharp angle is infinite. Consequently, plastic deformation occurs at very low pressures. This deformation evenly occurs under a force, as shown in Figure 13. The magnetic row bar and grinding oil produce gaps during the grinding process, for which there are three cases.

In the first case, the friction and removal rate are very small, and the lubrication type is dry friction. The film thickness is determined by the surface force between oil and plate. Assuming that the oil film thickness is t_0_, increasing the load pressure will cause large plastic deformation, and form a critical state. In the second case, the magnetic row bar is in contact with the oil film, but the pressure is completely borne by the nanodiamonds. Under these circumstances, the deformation is δ. For δ > δ_0_, the pressure continuously increases and is borne by the grinding oil and diamonds. The third case is common in precision grinding. Mixed lubrication is provided by the lubricating oil. The film thickness is denoted as T, the total roughness is denoted as RA, and the effective molecular diameter is denoted as RG. Here, the protrusion of the diamond is considered the surface roughness. In this case, the roughness satisfies Φ = δ + t for t/Ra < 1.

Yifei Mo [19] et al., investigated the ultra-precision grinding mechanism by performing large-scale molecular dynamics simulations with realistic force fields to establish the friction law of nanoscale dry contact: the friction force was demonstrated to linearly depend on the chemical interactions that occur during the contact process. We define the contact area as being proportional to the number of interacting atoms and show that the linear relationship between friction and the contact area observed on the macroscopic scale can be extended to the nanoscale. The model predicts that when the adhesion between contact surfaces decreases, the dependence of the friction force on the load changes from nonlinear to linear. Thus, the breakdown of continuum mechanics can be understood in terms of the multirough nature of the contact, and the asperity theory of friction can be applied to the nanoscale. Komvopoulos [20] et al., investigated the multiscale (fractal) surface roughness and the elastoplasticity and elasticity of hard abrasive nanoparticles embedded in a soft surface layer of a rigid polishing plate for nanoscale surface polishing by using a 3D stochastic model and a quasi-static mechanical analysis method. The influence of the embedding process on the surface morphology, roughness and grinding rate of the material was analyzed. The analysis method was used to calculate the steady-state roughness of the polished surface. The grinding of the material was modeled based on the surface contact pressure, polishing speed, original morphology and mechanical properties of the polished surface, average size and density of nanoparticles, and surface roughness of the grinding disc. Numerical results were obtained for the rate and wear coefficient. Cheung [21] et al., experimentally analyzed the material removal characteristics and surface morphology produced by valve-cover polishing: when the abrasive was clamped at the interface between the pad and the workpiece, wear mainly occurred via plastic deformation, and material was removed by abrasive particles. A multiscale material removal model and a surface generation method were established. An abrasive wear model and relative and cumulative removal process models for surface generation in bonnet polishing were established based on contact mechanics, kinematics theory, the abrasive wear mechanism, etc. A two-dimensional finite element model was used in conjunction with linear elastic fracture mechanics to analyze cracks induced by the sliding of rigid asperities. Variations in the roughness, interference depth, sliding friction, crack location and crack surface were found to affect the crack growth rate, direction and principal mode based on the maximum tensile range of the friction coefficient and shear stress intensity factor. Based on the results of these analyses, several experiments were repeated under improved conditions (i.e., the novel float method with an ultrafine slurry was used), and a better surface Ra was obtained. As shown in Figure 14b, we used an ultrafine slurry with an average diamond diameter of 100 nm instead of the normal-sized (approximately 118 nm) diamond slurry in Figure 14a.

The plate was subjected to precision grinding. The residual surface of the nanodiamonds on the plate was observed by a Keyence high-power microscope. The results are shown in Figure 15. The results show that the ultrafine slurry (Figure 15b) was more uniformly embedded in the plate than the normal slurry.

Atomic mechanics microscopy (AFM) was used to measure the surface roughness and pole-tip recession of the AlTiC substrate. The change in magnetic head resistance before and after precision grinding was measured, and the removal rate of the matrix material of the head was calculated. Figure 16 compares the surface produced under normal and improved conditions. The finest surface shown in Figure 16b produced under improved conditions is significantly smoother than the surface in Figure 16a with 0.165 nm Ra.

Figure 17a shows that after several repeated grinding experiments, the mean Ra can reach 0.197 nm under improved conditions but is only 0.263 nm under normal conditions. Figure 17b shows that the mean SF scratch depth can reach −1.009 nm under improved conditions but is only −1.228 nm under normal conditions.

## 4. Conclusions

Experiments were repeated several times, and the finest surface Ra obtained was 0.165 nm, without black spots or scratches. Thus, using a slurry containing small diamonds together with a novel float grinding process can reduce the surface roughness of a magnetic head and yield a smooth surface. In the improved grinding process, lubricant is only sprayed on the plate during grinding, and the nanodiamond particles form a mosaic on the surface of the plate, act as a grinding tool, and implement cutting and grinding. The small nanodiamonds (100 nm) in the aqueous slurry may reduce blackspots and scratches and subsequently have a stronger ability to efficiently stabilize the grinding of the magnetic head. According to the results of this study, the grinding plate prepared using small-sized nanodiamonds showed a very remarkable ability to reduce blackspots and scratches, which highlights the special value of this novel floating grinding process as a new method to control the surface roughness of magnetic heads and apply to the fine grinding of magnetic head applications.

## Figures and Tables

**Figure 1 nanomaterials-12-02763-f001:**
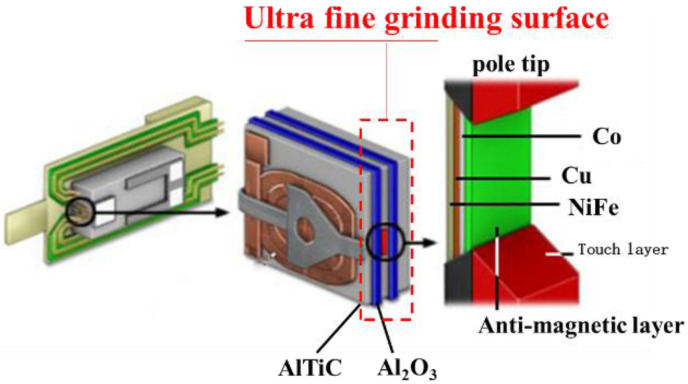
Illustration of an ultrafine grinding surface.

**Figure 2 nanomaterials-12-02763-f002:**
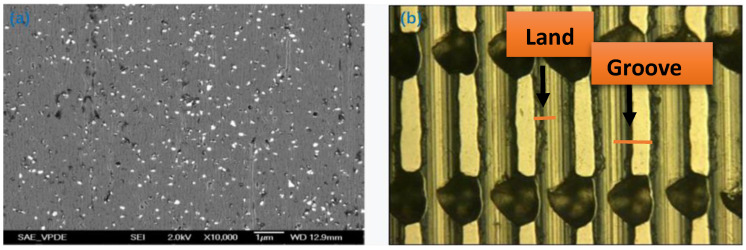
Abrasive distribution and Pit ratio of the tin plate: (**a**) abrasive distribution; (**b**) pit ratio check.

**Figure 3 nanomaterials-12-02763-f003:**
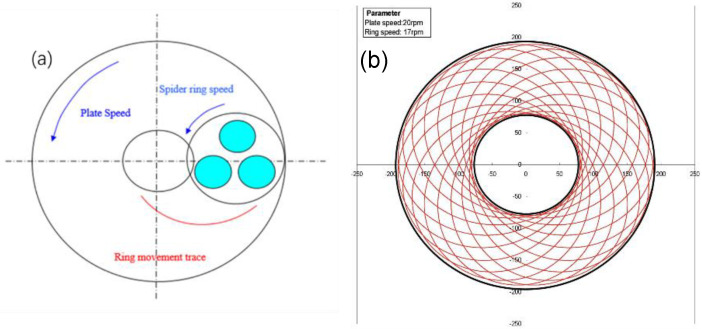
The scheme of the shaving process of the grinding ring: (**a**) motion model of grinding ring; (**b**) simulation diagram of the motion trajectory of the grinding ring.

**Figure 4 nanomaterials-12-02763-f004:**
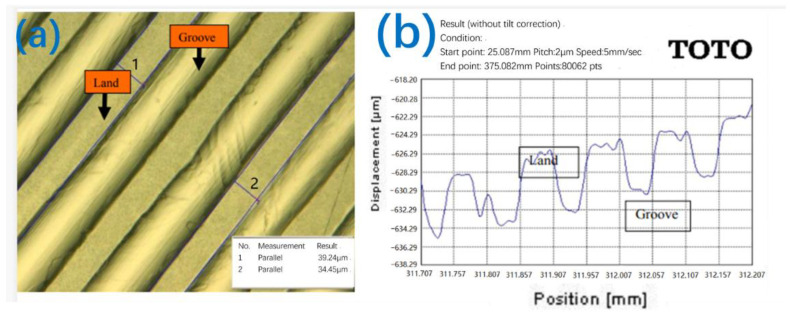
Observation of the surface topography of the lapping plate: (**a**) surface topography; (**b**) surface flatness.

**Figure 5 nanomaterials-12-02763-f005:**
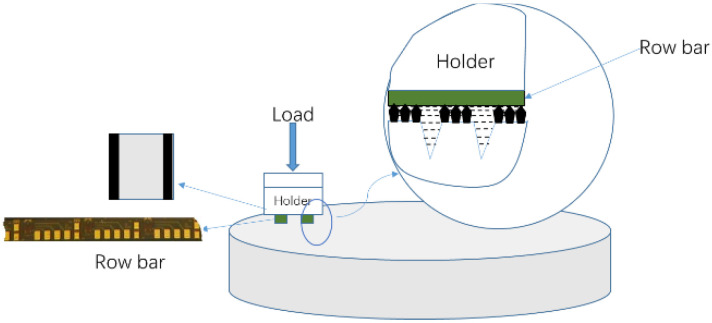
Schematic of the grinding process.

**Figure 6 nanomaterials-12-02763-f006:**
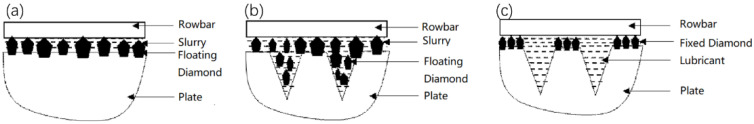
Comparison of different grinding methods: (**a**) traditional grinding process; (**b**) float grinding process; (**c**) novel float grinding process.

**Figure 7 nanomaterials-12-02763-f007:**
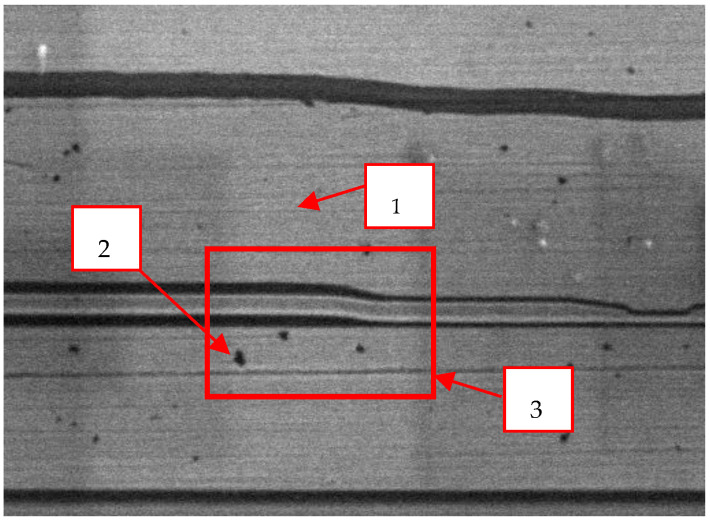
Secondary electron image of the pole area of the magnetic head (magnification: 40,000×).

**Figure 8 nanomaterials-12-02763-f008:**
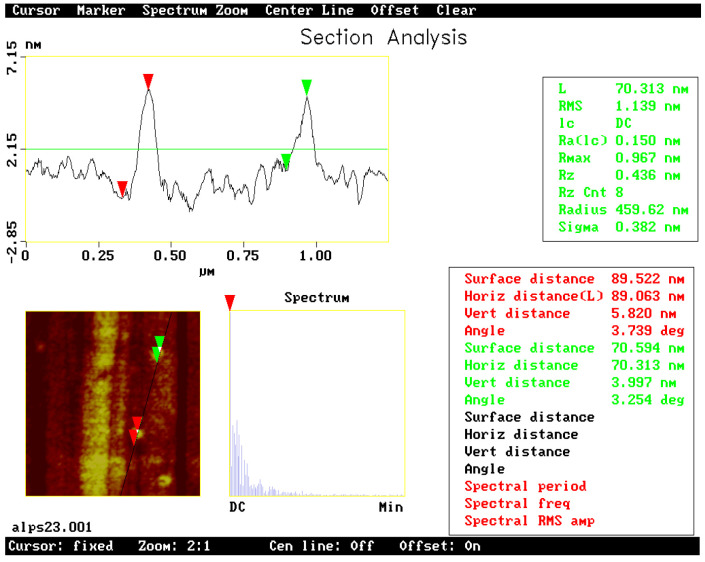
AFM image of the pole area of the magnetic head.

**Figure 9 nanomaterials-12-02763-f009:**
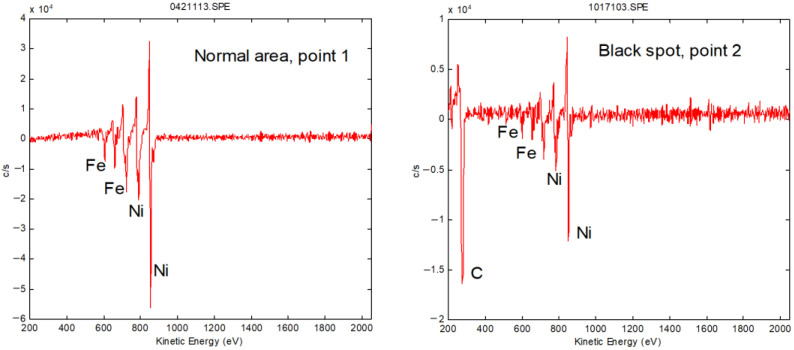
Auger electron spectra on normal area and black spot.

**Figure 10 nanomaterials-12-02763-f010:**
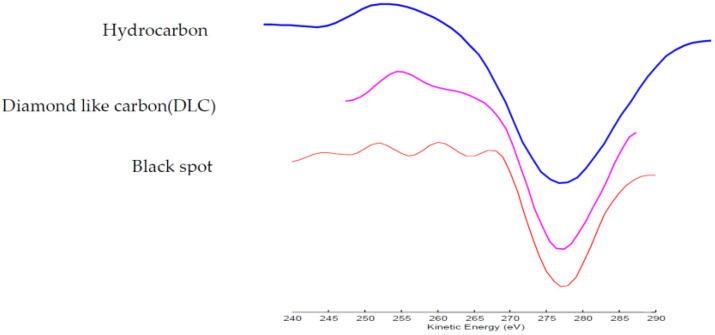
AES fine spectra of different types of carbon.

**Figure 11 nanomaterials-12-02763-f011:**
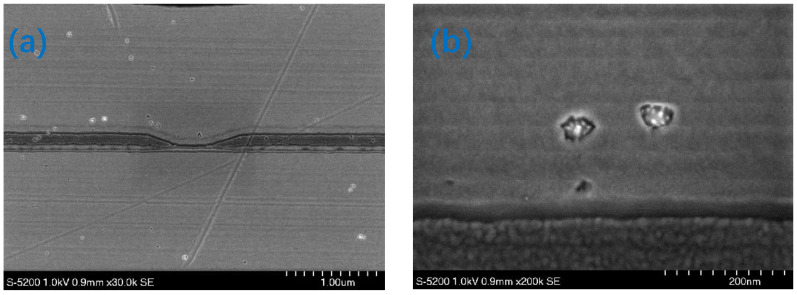
Super-SEM image of the black spots with different measurement range: (**a**) 1.00 μm; (**b**) 200 nm.

**Figure 12 nanomaterials-12-02763-f012:**
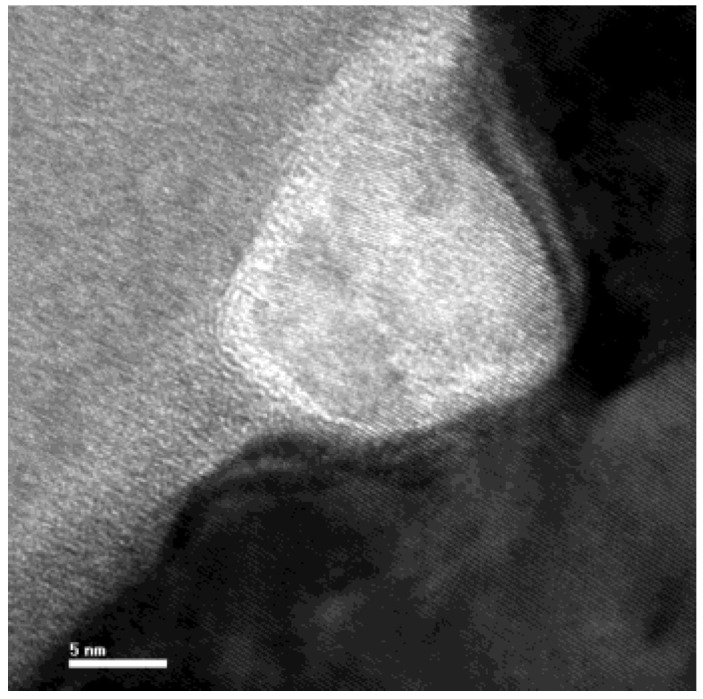
TEM images of a single black spot.

**Figure 13 nanomaterials-12-02763-f013:**
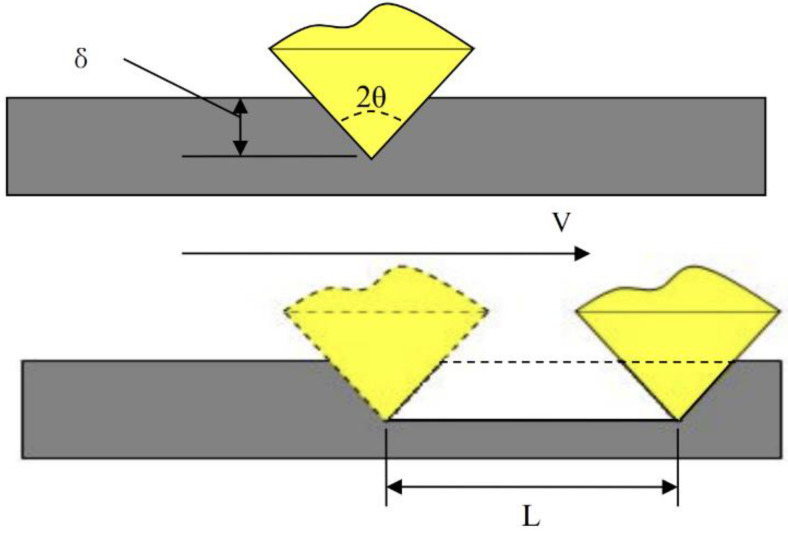
Mechanic for fine grinding.

**Figure 14 nanomaterials-12-02763-f014:**
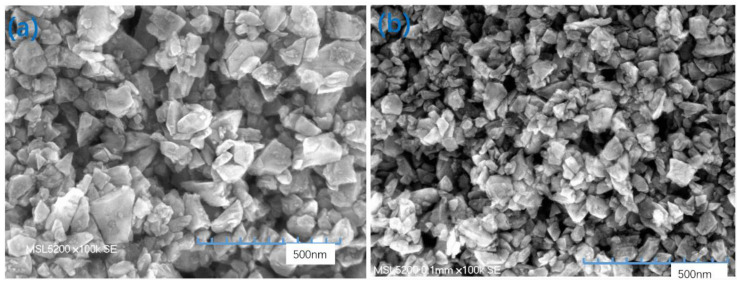
SEM images of slurries with diamonds of different sizes: (**a**) normal size (118 nm); (**b**) ultrafine slurry (100 nm).

**Figure 15 nanomaterials-12-02763-f015:**
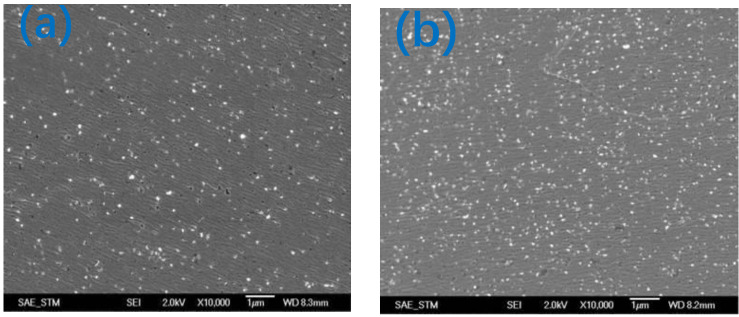
Keyence images of diamonds of different sizes embedded in a plate: (**a**) normal size slurry; (**b**) ultrafine slurry.

**Figure 16 nanomaterials-12-02763-f016:**
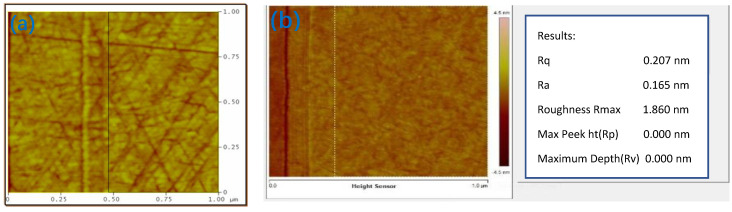
Comparison of AFM images of a slider surface produced under normal and improved conditions: (**a**) surface produced under normal conditions; (**b**) surface produced under improved conditions.

**Figure 17 nanomaterials-12-02763-f017:**
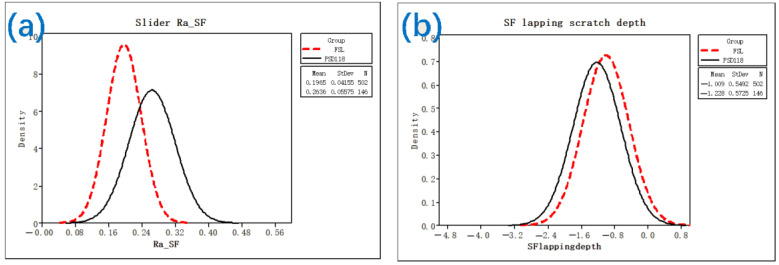
Comparison of Ra and the scratch depth under different conditions: (**a**) surface roughness (Ra); (**b**) scratch depth.

## Data Availability

Not applicable.

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
