# Peer review of "Using a Novel Floating Grinding Process to Improve the Surface Roughness Parameter of a Magnetic Head"

_nanomaterials, 2022, doi:10.3390/nano12162763_

Round 1

Reviewer 1 Report

This submission describes a novel grinding process to improve the Ra surface roughness parameter of a magnetic head. Its subject is interesting; however, some remarks should be done:

1.     The title should be corrected as “Using a novel floating grinding process to improve the Ra surface roughness parameter of a magnetic head”,

2.      Fig. 1 does not show the precision grinding process, but only a head image,

3.     The title of Fig.2 does not match its content,

4.     Cutting parameters as well as cutter features should be described,

5.     The Fig. 3 demonstrates the scheme of shaving process, not a dynamic model,

  1. The English should be corrected significantly. Bad language makes it difficult to understand the principles and results of the study.

Reviewer 2 Report

The author describes a fine polishing method that employs 100 nm diamond nanoparticles and a new grinding procedure, which allows to achieve a roughness level of 0.165 nm. The preparation of the polishing tool is described together with the polishing process and the analysis of the polished surfaces.

In my opinion the manuscript is poorly written so that interesting results could be obscured for a reader

However, the achieved Ra of 0.165 nm is quite remarkable so I believe that a modified version of the text could be accepted

Here some points that needs some modification

Abstract

“carbon has been found to be the main component of black spots and is also the main component of the diamonds in the slurry used for polishing.” It seems to say that carbon is the prevailing component in diamond, which doesn't make much sense

“The optimum conditions were found to be a pit ratio reach 30:70, a blank flatness (average) 1.8 μm.”

It would be better to define the  "pit ratio”. Also, the blank could be replaced with something like "steel plate".

Repetitions

line 37 “grinding the matrix of the magnetic head matrix”

line 45 “the metallic material of the magnetic head metal”

Unclear statements:

line 87 “The surface of the magnetic head is affected by processing”

in the paragraph the authors describe the realization of the abrasive tool, so it makes no sense to refer to the magnetic head out of the blue

Fig. 6 shows the improvement of the grinding technique, but there is no explanation on how this improvement is obtained. There are references to  “a micro damage model”  and “Ultrasonically enhanced intelligent control technology” which are not specified.

Fig. 16

The b) figure should be enlarged to read better the false color scale. There is no such scale in the a) figure

Lines 336-337

“Fig. 17(a) shows that the mean Ra can reach 0.165 nm under improved conditions but

only 0.263 nm under normal conditions.” Is there a reference to the “normal procedure” of which this submitted work represents further progress?

Conclusions

“According to the results of this study, the grinding plate prepared using small sized nanodiamonds showed a very interesting ability to reduce the blackspots and scratches, which highlights the roughness value of magnetic surface and for use in fine grinding applications.” Which is the subject of “hightlights?” How the last phrase “and for use…” is connected to the rest of the period?
